# An Empirical Study on the Relationship between Cognition and Metacognition in Technology-Enhanced Self-Regulated Learning

**Tuan Minh Tran [1,*]** and **Shinobu Hasegawa [2]**

1   School of Information Science, Japan Advanced Institute of Science and Technology,
    Nomi City 923-1292, Japan
2   Center for Innovative Distance Education and Research, Japan Advanced Institute of Science and Technology,
    Nomi City 923-1292, Japan; hasegawa@jaist.ac.jp
*   Correspondence: tran.minh.tuan@jaist.ac.jp

**Abstract:** Self-regulated learning (SRL) has become an indispensable ability for learners to succeed in self-study. A fast-growing number of learners worldwide opt for studying via technology-enhanced learning environments (TELEs) to enrich their education. Since the outbreak of the global COVID-19 pandemic, learners have relied more and more on online and distance learning for their own education purposes; this circumstance urges learners to self-regulate their learning processes. Striving for understanding learners' SRL ability in TELEs to provide them with sufficient support, this research analyzed the relationship between the two key factors of SRL, cognition and metacognition, from the context of self-study in TELEs. Applying our proposed hypothetical model on the relationship between cognition and metacognition, we conducted a pilot study in which 20 postgraduate students solved a complex academic task delivered via a TELE—the learning management system Moodle. In this experiment, the correlation between the students' cognitive and metacognitive scores was analyzed. The experimental results showed that there is a positive linear correlation between cognition and metacognition of a learner when he or she performs a complex task in TELEs, and such a correlation can be classified into different profiles. Implications include opportunities to help learners understand their SRL profiles and provide recommendations for further research on the granularity of SRL characteristics.

**Keywords:** cognition; cognitive strategies; metacognition; metacognitive strategies; self-regulated learning; technology-enhanced learning environments

## 1. Introduction

Self-studying via technology-enhanced learning environments (TELEs) is becoming increasingly popular, and it is a promising approach to sustainable life-long learning. Over the last two years, the COVID-19 pandemic has disrupted traditional learning routines, forcing almost all educational activities to transition to online environments, thus turning online learning into an unavoidable reality and a necessity. To self-study effectively, learners must have the ability to master their learning processes. This ability is termed self-regulated learning (SRL).

SRL is an active learning process by which learners approach and observe knowledge consciously, actively, strategically, and tactically with concrete goals in mind [1]. A good aspect of self-studying in TELEs is that these environments are ideal for learners to self-regulate their learning, as recent research has discovered [2]. Learners, when learning in a self-regulating manner, perform various activities both cognitively and metacognitively which are hardly fulfilled in traditional learning settings with the constraints of physical learning locations, timeframes, and facilities. However, these constraints are easily bypassed in TELEs.

TELEs provide various tools to support a wide range of online learning activities such as forum or message or chat for discussion and collaboration, videos or ebooks or animation as learning materials, quizzes or assignments for assessment, and wiki or note-taking for planning and reflection [3]. Therefore, they offer learners more autonomy in terms of place, time, pace, and controllability over learning resources than traditional learning settings. Such conditions enable learners with the SRL ability to self-study effectively, enjoyably, and with motivation [4].

Although they are offered similar supporting tools in TELEs, learners who lack the SRL ability do not self-study effectively. Besides being affected by motivation loss or lacking background knowledge or skills for online learning [5], these learners have also encountered anxiety when studying online because they are used to social aspects in traditional learning contexts but feel that are missing those in online learning contexts [6]. In addition, although they possess sufficient cognitive skills, learners weak in metacognitive skills such as self-discipline, self-monitoring or controlling, or self-motivation encounter difficulties when they self-study online [4]. Thus, the SRL process in TELEs exposes a certain relationship between the cognition and metacognition of learners. Cognition and metacognition seem to correlate linearly in some instances and causally in others, yet also to diverge under other circumstances. Recognizing such a relationship assists learners in adjusting and augmenting their SRL ability properly.

Discerning the challenges posed by the current learning circumstances and the opportunities enabled by TELEs to solve those issues, this paper aims to analyze how learners' cognition and metacognition relate when they self-regulate their learning in TELEs. The analysis was carried out by conducting a pilot study that followed the authors' hypothetical model about the relationship between cognition and metacognition [7]. We based our model on Zimmerman's SRL cyclical model, revised Bloom's taxonomy, and Flavell's metacognition theory. Built on these stable bases, the proposed model strives to bring out a sustainable learning approach by supporting learners' understanding of their cognitive and metacognitive strengths and weaknesses and assisting course designers in developing content that enables learners to self-regulate their learning.

The structure of this paper is organized as follows. In the next section, the related work, we describe SRL in TELEs, cognition and metacognition, and revised Bloom's taxonomy. Then, we begin the materials and methods section by describing our hypothetical model based on which the empirical study was conducted. Next, we illustrate the experiment design and delivery and discuss the results. Finally, the paper ends with implications and suggestions for further research.

## 2. Related Work

### 2.1. Self-Regulated Learning in TELEs

It is natural to see people reading a book yet not understanding much of the book for the first time, then adjusting their way of approaching that book, for instance, by taking notes, highlighting paragraphs, asking questions, and wondering, so that they can get more out of the book from the second or third reading. A similar process occurs when people perform other cognitive activities such as studying a new lecture or doing a course assignment. This ability to adjust oneself to know a subject matter better is known as self-regulated learning (SRL). Professor Barry Zimmerman [8] (p. 541), a leading researcher in SRL, maintains that "Self-regulated learning involves metacognitive, motivational, and behavioral processes that are personally initiated to acquire knowledge and skill such as goal setting, planning, learning strategies, self-reinforcement, self-recording, and self-instruction". SRL is an innate ability [9], and we are more aware of SRL when we grow up and are told about and guided through it. There has been a rapid growth of research in SRL, resulting in models illustrating elements and processes of SRL as well as intervention methods to help people practice SRL [10]. At the present time, as learning expands outside of traditional classrooms into online environments, SRL in TELEs has been attracting increased attention.

TELEs are facilities enabled by information and communication technologies (ICT) to support learning processes. The support of TELEs spreads widely from learning content storage and delivery, academic self-assessment, virtual social interaction to individual learning adaptation, virtual assistance, and so forth. With the significant breakthrough in information technology and computer science in the last two decades, support from TELEs has become available for thousands of learners worldwide. Two popular forms of TELEs are learning management systems (LMS) and online learning platforms. Some well-known TELEs are Blackboard, Moodle, and various massive open online course (MOOC) platforms such as EDX, Coursera, and MIT OCW, which deliver thousands of academic courses.

One of the benefits learners have from TELEs is that learners are liberated from space and time constraints for studying. Another benefit is that TELEs enhance every aspect of SRL [3], including but not limited to setting goals, planning, organizing, monitoring, and self-evaluating. For instance, John C. Nesbit et al. [11] introduced a cognitive activity support software named gStudy to assist learners in SRL and recorded the learners' activity traces for research and support purposes. Green and Azevedo [12] concluded that the combination of students' monitoring ability and use of hypermedia, which are kinds of learning resources in TELEs, plays an important role for them to learn complex subjects. Support to SRL from TELEs extends to an ability to remind learners of steps in SRL processes and assist them in performing steps such as planning, monitoring, and adapting learning activities [13]. About the societal aspect, TELEs provide tools to support learners in communicating with their peers and instructors and adaptively changing content to meet individual learning needs.

Furthermore, with the advances in ICT, machine learning (ML) techniques and algorithms enable TELEs to know individual learners' ways of learning so as to adapt contents and supportive mechanisms in much diverse and personal manners. As illustrated in Figure 1, the role of ML in TELEs to support SRL may work in this way. SRL starts a learning process in TELEs by having learners use features offered by TELEs to self-study a certain learning task. The way the learners apply the facilities to their learning processes and the learners' corresponding progress are then fed into ML components of TELEs for analysis and training ML to know more about the learners' learning patterns. After the analysis, TELEs identify a combination of facilities and the usage frequencies of the features that might help the learners progress. TELEs use these types of information to know the learners' learning patterns and then adjust themselves to suggest other features that the learners miss but should use to achieve better learning outcomes.

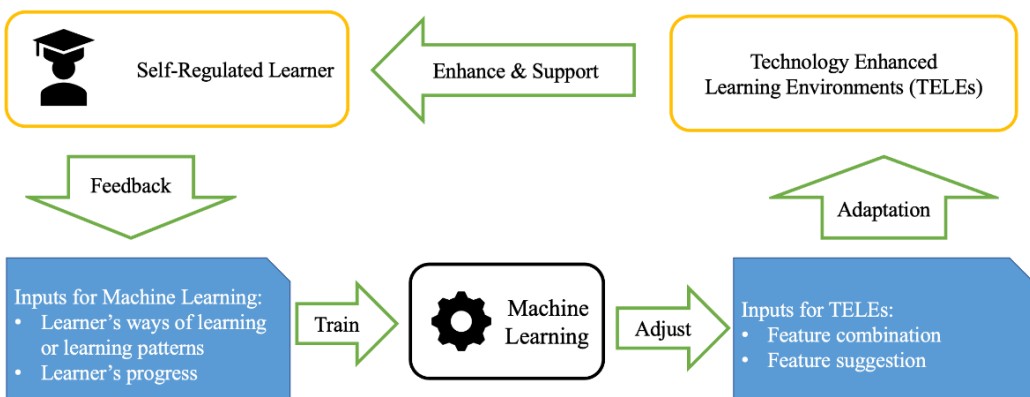

**Figure 1.** The relations between SRL, TELEs, and ML.

Despite various supports from tools and mechanisms, there are still learners who are neither effective nor even, in worse cases, desperate at self-study. Barak et al., in a survey [4], showed that either learners lacking in SRL ability or those acquainted with direct instructions might encounter difficulties and miseries when learning online. The autonomy and facilities offered by TELEs might confuse learners that they cannot determine a proper learning path among several options. It seems to be the ability to set an appropriate course

of activities in TELEs that enables learners to take advantage of facilities provided by TELEs and to self-study effectively. Empirically, the survey above delivered by Barak et al. [4] indicated that online learners with metacognitive skills of self-discipline and cognitive strategy regulation could use resources provided by TELEs effectively. Therefore, they tend to self-regulate their learning well in TELEs. On the other hand, other students without metacognitive skills, although they use cognitive strategies, seem not to self-study as well in TELEs as their SRL peers. Neither metacognition nor cognition alone seems to contribute to achievements in SRL in online learning. Hence, a relationship between these two factors seems to exist for the explanation of effective SRL in TELEs.

*2.2. Cognition and Metacognition*

Learning involves two objects: the subject matter to be learned and the learning process itself. Learners apply cognitive strategies to the subject matter and simultaneously apply metacognitive strategies to observing the whole learning process either consciously or unconsciously. It is in the nature of the mind that learning happens in such a way [9]. Flavell, the father of metacognition research, defined metacognition as one's knowledge and awareness about one's cognition [14]. Metacognition is in action whenever we work on a cognitive task such as studying or solving problems. The more complicated the task is, the more clearly metacognition is revealed. According to Flavell [15], metacognition comprises two aspects: (i) metacognitive knowledge, which is threefold knowledge about a task we work on, a person related to the task, and cognitive strategies applied to that task; and (ii) metacognition experiences, which support the regulation and selection of strategies to apply when working on the task.

Figure 2 illustrates metacognition according to Flavell's metacognition model. For instance, when a person works on a cognitive task such as learning a new English grammar lesson, his metacognition works according to the following procedure. The metacognitive knowledge accumulates three types of information which are (a) the person himself, his strengths and weaknesses in language learning, (b) the task and related goals such as re-membering grammar syntax and being able to apply it in real-life contexts, and (c) cognitive strategies to learn this grammar lesson. The cognitive strategies are accumulated in that person's cognitive repertoire of learning foreign languages. Those strategies could be read-ing grammar syntax, doing grammar exercises for deep remembering, and then practicing the grammar syntax from memory instead of with written text to reinforce knowledge effectively. During the learning process, his metacognition keeps monitoring and checking progress against the planned goal. He might find the current course of cognitive strategies does not perform as well as planned. At this time, the metacognitive experience revises and updates metacognitive knowledge and issues alternative cognitive strategies to steer him toward the planned goal.

Understanding how the mind works, specifically how cognitive strategies perform and how to regulate those strategies in an academic context, is beneficial for improving learning [16] in general and improving SRL in particular. There are several studies about links between cognition and metacognition, mostly toward academic performance. Most of the studies agreed that the better the cognitive and metacognitive strategies are, the better the academic performance is [17,18]. As Winne [19] puts it, SRL is a complex intertwinement of cognition and metacognition. This relationship is difficult for learners to be aware of; even when recognizing such a relationship, learners often find it vague to identify its characteristics and quality in corresponding to the learners' SRL levels. Thus, it would be informative and beneficial for learners' understanding of their SRL profiles for them to have a checklist of characteristics from the relationship between cognition and metacognition, how effectively they self-regulate their learning and what appropriate improvement they should aim at. The experimental results in this paper seek to answer those points.

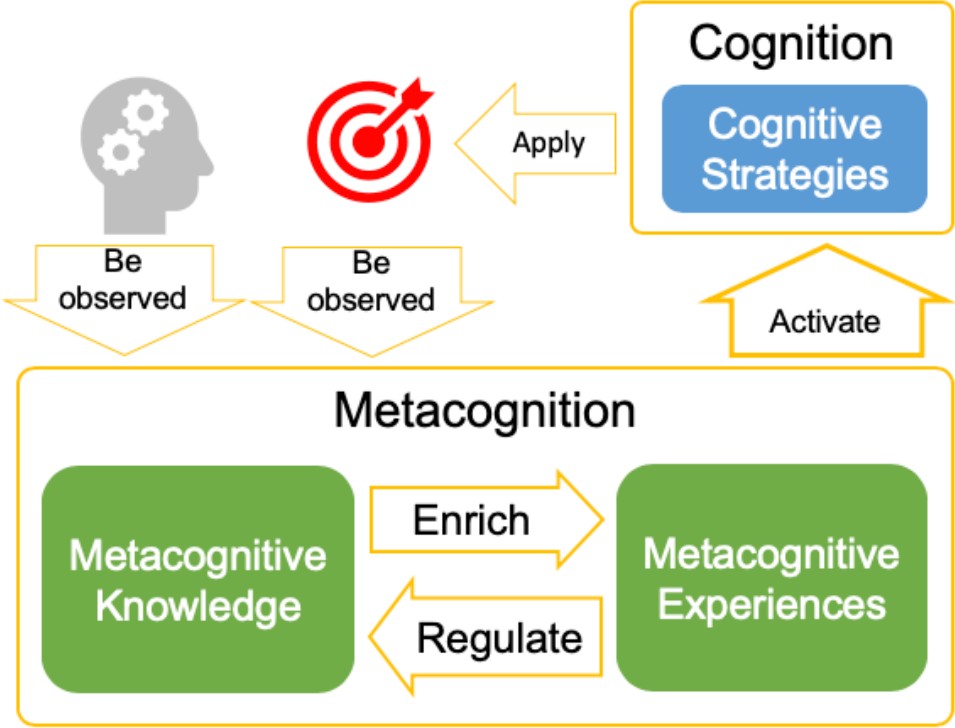

**Figure 2.** Components of metacognition according to Flavell's metacognition model.

### 2.3. Revised Bloom's Taxonomy

Bloom's taxonomy is a popular framework for identifying and classifying learning objectives in order to determine learners' progress in knowledge attainment. The revised Bloom's taxonomy from 2001 refines the description of learning objectives in a finely detailed manner. According to the revised Bloom's taxonomy [20], a learning objective has two dimensions: a knowledge dimension and a cognitive process dimension. The knowledge dimension is about what a learner knows; it categorizes knowledge into four types: factual, conceptual, procedural, and metacognitive types of knowledge. Knowledge is categorized in a hierarchical complexity, from concrete elemental facts through abstract ideas to guidance for actions. The cognitive process dimension is about what a learner can do with his or her obtained knowledge, and it classifies cognitive activities into six levels which are Remember, Understand, Apply, Analyze, Evaluate, and Create. The cognitive process becomes more and more complex as learners act from simply remembering or understanding information, through trying to apply what they have learned and finally to being able to synthesize a body of knowledge to create new products. In summary, a learning objective comprises knowledge one has learned, one's ability to bring out values from the knowledge, and interestingly one's awareness of one's learning process and ability to regulate such a learning process. That is to say, the revised Bloom's taxonomy is designed to classify levels of a combination between the cognition and metacognition of a learner. With a hierarchical arrangement of learning objectives, the taxonomy can help design learning tasks whereby learners can reveal their SRL levels in the form of the relationship between cognition and metacognition in their learning process and their performance.

### 3. Materials and Methods

We proposed a hypothetical model [7] for evaluating the quality of the relationship between the cognition and metacognition of learners in self-regulated learning in TELEs. Moreover, in this research, we delivered a pilot study to validate the model. We begin this section with a brief description of the model and then illustrate the pilot study.

### 3.1. Hypothetical Model of a Relationship between Cognition and Metacognition in Technology-Enhanced Self-Regulated Learning

This hypothetical model was developed on the threefold foundation of the SRL cyclical model of Zimmerman [1], the revised Bloom's taxonomy [20], and Flavell's metacognition theory [15]. It hypothesizes that one's metacognition and cognition expose a positive correlation via one's performing complex learning tasks in TELEs. In self-regulated learning with a certain complex task, those who perform effective metacognitive strategies will probably attain high cognitive performance; conversely, those with high cognitive performance are likely proficient at metacognitive strategies. That positive correlation is measurable by the correlation between cognitive and metacognitive scores, then classified into SRL profiles—descriptions of SRL characteristics according to SRL maturity levels.

Let us describe each aspect of the hypothetical model and how it demonstrates the cognition and metacognition relationship.

Zimmerman's SRL cycle model shows SRL as a process of three phases: (i) a forethought phase including processes for planning metacognitive, motivational, cognitive, and environmental elements for a learning purpose; (ii) a performance phase containing subprocesses by which learning activities and monitoring and controlling activities of learning are carried out; and (iii) a self-reflection phase containing mostly metacognitive behaviors to assess the learning process and performance. Each phase of the SRL process is a composition of interoperations between cognition and metacognition whose mechanism is validated by Flavell's theory and whose outcomes can be recorded as the learners' learning behaviors in revised Bloom's taxonomy.

How does the relationship between cognition and metacognition reveal itself in such a way? We can start from our experiences when we learn a complex and challenging subject matter.

To complete such a learning task successfully, we would begin the learning process with diligent observation of the task, looking at it from various angles, determining a certain degree of achievement in consideration of our current background knowledge, then selecting a suitable environment in which we can learn efficiently and choosing appropriate cognitive strategies for learning the task. That is the kind of plan we would make for studying complex learning tasks. After planning in detail, the next step is to learn—to put our plan into action. Not only do we apply cognitive strategies for learning and observing bit by bit the knowledge delivered by the task, but in the meantime, we also observe the way we learn. We monitor whether our learning performance is progressing as well as planned and whether certain cognitive strategies or certain learning goals should be adjusted. For instance, whether the planned goal was unreachable within the current time constraint; if so, we should cut off some objectives or spend extra time rehearsing a certain learning point to consolidate it in our long-term memory. Finally, when completing the learning task, we evaluate outcomes that we have learned, checking if we have mastered the knowledge from the task against the goal. In addition, we reflect on our way of learning to see which cognitive strategies are effective for such a task so that we can polish our own learning pattern and approach similar tasks in a similar manner in the future.

Looking at how we learn, we can see that our cognition and metacognition shape and adjust and drive each other. The more we are aware of our cognition and metacognition, the clearer we see their collaboration when we learn; and the more we pay attention to such collaboration when we learn, the better it gets; and as a result, we can learn in a more self-regulated manner.

The hypothetical model shows that the relationship between the cognition and metacognition of learners is revealed when they learn complex tasks in a self-regulated way. A complex task [2] satisfies the following four requirements: (1) containing high-level educational behaviors in revised Bloom's taxonomy, (2) performed through a multiple-step process, (3) having outcomes requiring investigations to see them clearly, and (4) requiring a diligent plan to perform. The relationship between cognition and metacognition is established via the correlation between the cognitive score that learners earn from completing

the task and the metacognitive score that learners reflect on their strategies applied to doing the task by answering a metacognitive questionnaire after finishing the task.

How are cognition and metacognition scored? As working on a learning task is a multi-step process following Zimmerman's cyclical SRL model, learners earn several partial cognitive scores for their completion of each of the steps in the process, and these partial cognitive scores are accumulated into a final total cognitive score. After the learners have completed the learning task, they are requested to answer a Metacognitive Awareness Inventory (MAI) questionnaire [21] to reflect on their learning process. This step is also the third stage in Zimmerman's cyclical SRL model. Their metacognitive answers are scored on Likert scales and then accumulated to form a final total metacognitive score.

In the following section, we present the empirical study in which we followed this model to conduct an experiment about the relationship between the cognition and metacognition of learners when they learn on Moodle in a self-regulated manner.

### 3.2. Experiment Design

In this paper, we conducted a one-group post-test-only experiment for a pilot study to analyze the cognition and metacognition correlation trajectory in order to test our hypothetical model described above. The experiment results would be evidence to adjust the model and determine whether a larger study is required. An experiment was designed as follows. An experimental group of graduate students joined an approximately two-hour assignment delivered via the Moodle system. As the assignment was delivered online, the students worked on the assignment separately at a convenient place and time of their choice. While the students were working on the assignment, their behaviors relating to their working process were stored in the system to the extent of access to resources, attempts on quizzes, quiz scores, and source code writing. After finishing the assignment, the students answered a metacognitive questionnaire. Their answers to the metacognitive questionnaire and the final results from the assignment were analyzed and matched to demonstrate the relationship between their cognition and metacognition in this experimental learning process.

### 3.3. Task Design

To meet the task criteria, we chose the computer science area in which the task assignment was designed. The task was in machine learning because this context fulfilled all the criteria mentioned above, which is required to reveal the relationship between cognition and metacognition. The experimental task was applying existing machine learning algorithms and calibrating their parameters to build a high-accuracy handwritten character recognition model based on the EMNIST [22] dataset of handwritten characters.

This experimental task meets all of the required criteria:

- It researches the complex levels 4 (Analyze), 5 (Evaluate), and 6 (Create) in revised Bloom's taxonomy;
- It requires the following steps: (1) analyzing and understanding dataset structure, (2) comprehending selected algorithms, (3) training a prediction model, (4) testing the prediction model on new data, and (5) calibrating parameters to achieve better accuracy;
- The optimal achievable accuracy is not specified;
- Performing this multi-step task requires an ordered course of cognitive actions and progress monitoring.

For this task the participants were expected to meet the following objectives, ranging from concrete and straightforward levels to complex and abstract levels in revised Bloom's taxonomy. Working on such a task, learners were encouraged to exploit their SRL skill set. Table 1 shows the learning objectives from this task.

For this task the participants were also expected to have the following prerequisite knowledge and practice in the machine learning area:

(1)    Prior knowledge of feature selection, training algorithms for image recognition, parameters for training algorithms, loss functions, Python machine learning libraries;

(2)  Experience with the procedure for developing a machine learning prediction model;

(3)  Ability to perform activities such as understanding data (data structure, feature selection), pre-processing data (scaling, normalizing), algorithm evaluation metrics selection (selecting loss functions), algorithm tryout (training models, applying cross-validation, modifying parameters), algorithm and model comparison (comparing loss function results), algorithm and model selection (presenting selected algorithms, accuracy, standard deviation), algorithm optimization, and prediction optimization (calibrating parameters, combining multiple models).

**Table 1.** Revised Bloom's taxonomy-based learning objectives from the experiment task.

| Revised Bloom's Taxonomy | Objectives |
| --- | --- |
| Remember | Recognize the type of ML problem to be solved<br>Recall available ML algorithms that are suitable for solving this problem<br>Identify the structure of the EMNIST dataset<br>Recall ML tools and libraries which are possibly used for this problem<br>Recall the metrics for evaluating prediction model accuracy |
| Understand | Distinguish different types of ML problem<br>Distinguish ML training algorithms and prediction model<br>Distinguish train, development, and test dataset<br>Describe ML training algorithms and their usage purpose<br>Describe procedures to build a prediction model |
| Apply | Implement ML algorithms using available libraries to develop prediction models |
| Analyze | Organize dataset into a suitable partition to achieve efficient and effective training performance<br>Modify parameters of ML training algorithms to improve their performance |
| Evaluate | Compare the performance among ML training algorithms using appropriate metrics |
| Create | Plan a procedure using libraries to execute ML training algorithms on the datasets<br>Build a prediction model from the dataset |

*3.4. Criteria for Participants*

In this pilot study, an experiment was conducted in an international institution for postgraduate education in Japan in which potential students for the experiment have certain SRL abilities and have studied courses related to the experimental task.

In this experiment, we measured the relationship between cognition and metacognition by Pearson's correlation coefficient. Since the hypothetical model hypothesizes a positive correlation between these two factors, we expected the Pearson's correlation coefficient to be positive at a significant level of 0.05. The results would help to determine whether further experiments with a large sample size are needed. Thus, as suggested by Hertzog [23] for determining sample size for such expectations, we recruited 20 participants for this study.

Participants were second-year master's students and first-year doctoral students from 24 to 28 years of age who had completed or were studying in machine learning courses. When doing the experimental task, these participants would demonstrate a wide range of self-regulated learning skills. They would review prior knowledge in machine learning courses, remind themselves about experiences of similar tasks, set specific goals from the task analysis, and plan a series of activities to perform the task.

The participants were expected to go through the following flow:

- Problem set reading comprehension: The participants read the problem statement for comprehending tasks that needed to be done;
- Planning (SRL—forethought phase): The participants planned expected outcomes, timeline, which steps to follow, and what methods to apply to achieve the outcomes required in the problem set;
- Preparation (SRL–forethought phase): The participants prepared facilities and materials for working;

- Review (SRL—performance phase): The participants reviewed previous knowledge which they found necessary for working on the problem set;
- Performance (SRL—performance phase): The participants worked on the problem set;
- Self-assessment (SRL—evaluation phase): The participants evaluated the final results and answered the MAI questionnaire.

At each step in the flow, the participants were expected to read instructions and do quizzes. These activities were signs that the participants demonstrated cognitive strategies, and the participants also earned cognitive scores for each quiz they had completed. At the final step, the participants answered a metacognitive questionnaire whose questions reflected the participants' whole learning process through the task.

*3.5. Moodle as a TELE*

The experiment was conducted via Moodle. Moodle [24] is one of the most popular open-source online learning environments. Moodle can be used in blended learning or online learning. Providing the extension capability, Moodle allows for integrating external programs in the form of plugins to extend Moodle's functionalities in various manners. Recently, Moodle introduced a machine learning extension [25] which supports the opportunity to develop machine learning models serving more diverse needs of online learning.

Besides the mechanism for adding new functionalities, Moodle itself contains many tools supporting Bloom's taxonomy-based cognitive behaviors [26], which are helpful for this research and further study into the relationship between cognition and metacognition in TELEs. Specifically for this research, we used the following Moodle features: quiz with various question types to support planning and preparation work, video and ebook resources to support knowledge revising, wiki to support note-taking, and assignments to support final work submission and assessment.

*3.6. Experiment Delivery*

The experiment content was delivered on the Moodle platform as follows.

1. Introduction. This section presented the purpose of the experiment and the expectation for the participants' behaviors when joining the experiment. The introduction on the experimental Moodle site was written as in Figure 3.

The experiment is conducted as follows:

> *You will solve a Machine Learning task within a limited given time. During the process, you are encouraged not to jump right into dealing with the task but to spend some time diligently reading the task description, recalling your prior knowledge, doing a series of short quizzes that supports you on this task, and then solve the main task. This experiment takes approximately about 2 hours but you don't have to do everything all at once. The only thing that matters is to complete the task since it is necessary for the verification and validation of the hypothesis model.*

Again, thank you for joining the experiment. Wishing you joy and success.

Let's start the experiment. Please follow the procedure below to work through the experiment.

**Figure 3.** Snapshot of experiment introduction section on the experimental Moodle site.

2. Experiment procedure. This procedure suggested the order the participants should follow in order to achieve a good result when solving the problem in the experiment.
3. Task description. In this part, the problem that needed to be solved was described. In addition, resources for supporting the participants in solving the problem were provided. The resources included the runnable source code that the participants could use as a starting point and a quiz that the participants could complete to prepare their knowledge for solving the problem.
4. Source code reading instruction. This section guided the participants through the source code to help them understand how it runs and where they could modify it.

5.  Task comprehension. This section provided the participants with a review quiz to help remind them of the knowledge necessary to solve the problem.
6.  Preparation. This section contained a quiz to remind the participants of specific machine learning knowledge that the problem belongs to so that the participants could focus on applying those specific knowledge domains for this problem.
7.  Self-assessment. This section contained a quiz that asked the participants to reflect on their progress through the task to distill their achievement from the task.
8.  References. This section included lessons about machine learning knowledge related to this problem for participants to consult.
9.  Questionnaire. This final section contained a subset of 19 items from the Metacognitive Awareness Inventory (MAI) questionnaire. This MAI subset provides a consistent, invariant, and good fit for students' metacognition [27].

Quizzes in Lists 5–9 were to assist students in expressing their metacognition into visible activities. The participants were free to follow the experiment in any order.

The experiment scoring structure was comprised of (i) cognitive score, which was accumulated by the main task score together with quizzes scores, and (ii) a metacognitive score from answers for the MAI questionnaire. The highest cognitive score was 100, in which the main task was 50 scores at maximum, four quizzes were of 10 scores each at maximum, and 10 scores for finishing the MAI questionnaire. The reason for the addition of 10 scores for finishing the MAI questionnaire was that answering this questionnaire is a manifestation of the self-assessment phase in the SRL cyclical model. It is a necessary phase to close an SRL process. The highest metacognitive score was 95, which accumulated scores from nineteen 5-level-Likert-scale MAI items [27].

## 4. Results

Three main outcomes from the experiment are recorded, including the (1) accuracy of the prediction model, which was the main outcome from solving the experiment problem, (2) the accumulative cognitive score, which was the total score of each participant's performance on the experiment problem and quizzes, and (3) the metacognitive score from the MAI questionnaire. The accuracy of the prediction model is excellent if it is over 90 percent, while it is poor if below 80 percent.

The bar chart in Figure 4 illustrates a correlation between the prediction model accuracy and the cognitive score and the metacognitive score of each participant in descending order according to cognitive scores. Overall, the students who earned the highest cognitive scores achieved prediction models with the highest accuracy and also had high metacognitive scores; however, those with the lowest cognitive scores also had high metacognitive scores.

Table 2 and Figure 5 show results from the test and histograms for normal distribution over the collected data. Preliminary analysis showed the relationship to be linear with both cognitive score and metacognitive score normally distributed, as assessed by Shapiro–Wilk's test ($p > 0.05$).

A Pearson's product-moment correlation was run to assess the relationship between metacognition and cognition in the participants. A scatterplot of cognition (cognitive scores) against metacognition (metacognitive scores) was plotted. Visual inspection of the scatterplot, demonstrated in Figure 6, indicated a linear relationship between the variables with several outliers, which is discussed later in the Discussion section.

**Table 2.** Test of normality result.

|  | Shapiro–Wilk | | |
| --- | --- | --- | --- |
|  | Statistic | Degrees of Freedom | *p*-Value |
| Cognitive score | 0.930 | 20 | 0.152 |
| Metacognitive score | 0.936 | 20 | 0.204 |

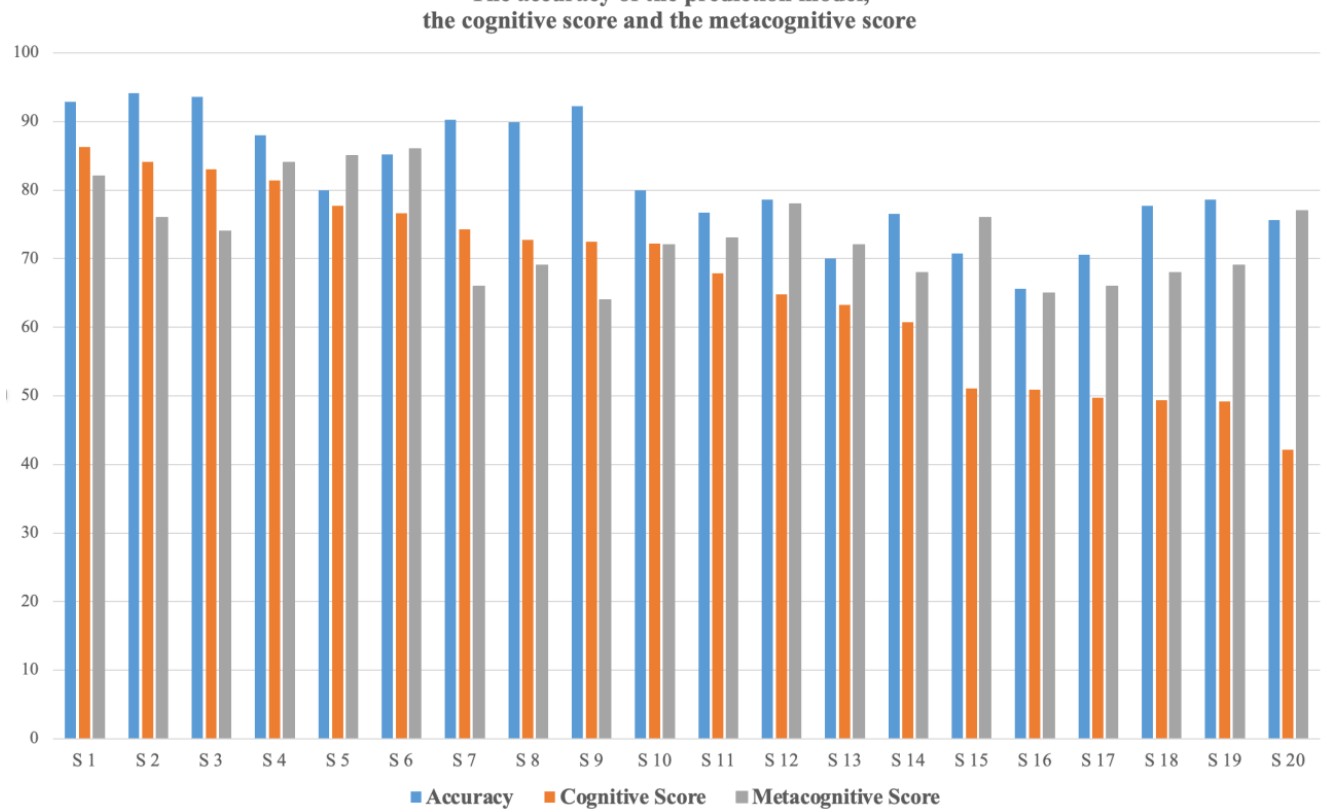

**Figure 4.** Bar chart of the accuracy, cognitive scores and metacognitive scores that students earned in the experiment.

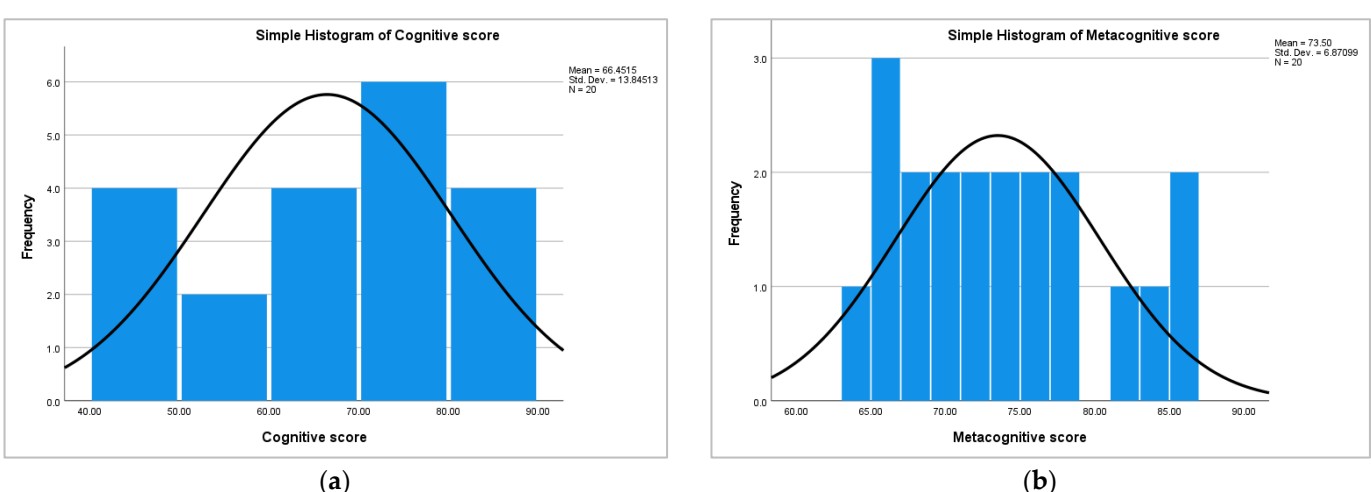

(**a**)            (**b**)

**Figure 5.** (**a**) Histogram of cognitive scores: mean = 66.45, SD = 13.84; (**b**) Histogram of metacognitive scores: mean = 73.50, SD = 6.87.

There was a statistically significant, moderate positive correlation between metacognitive score and cognitive score, r(18) = 0.449, $p < 0.05$, as shown in the test result in Table 3, with metacognition explaining 20% of the variation in cognition.

**Table 3.** Pearson's correlation test result.

|  |  | Cognitive Score | Metacognitive Score |
|---|---|---|---|
|  | Pearson Correlation | 1 | 0.449 * |
| Cognitive score | *p*-value (2-tailed) |  | 0.047 |
|  | N | 20 | 20 |
|  | Pearson Correlation | 0.449 * | 1 |
| Metacognitive score | *p*-value (2-tailed) | 0.047 |  |
|  | N | 20 | 20 |

* a statistically significant, moderate positive correlation between metacognitive score and cognitive score.

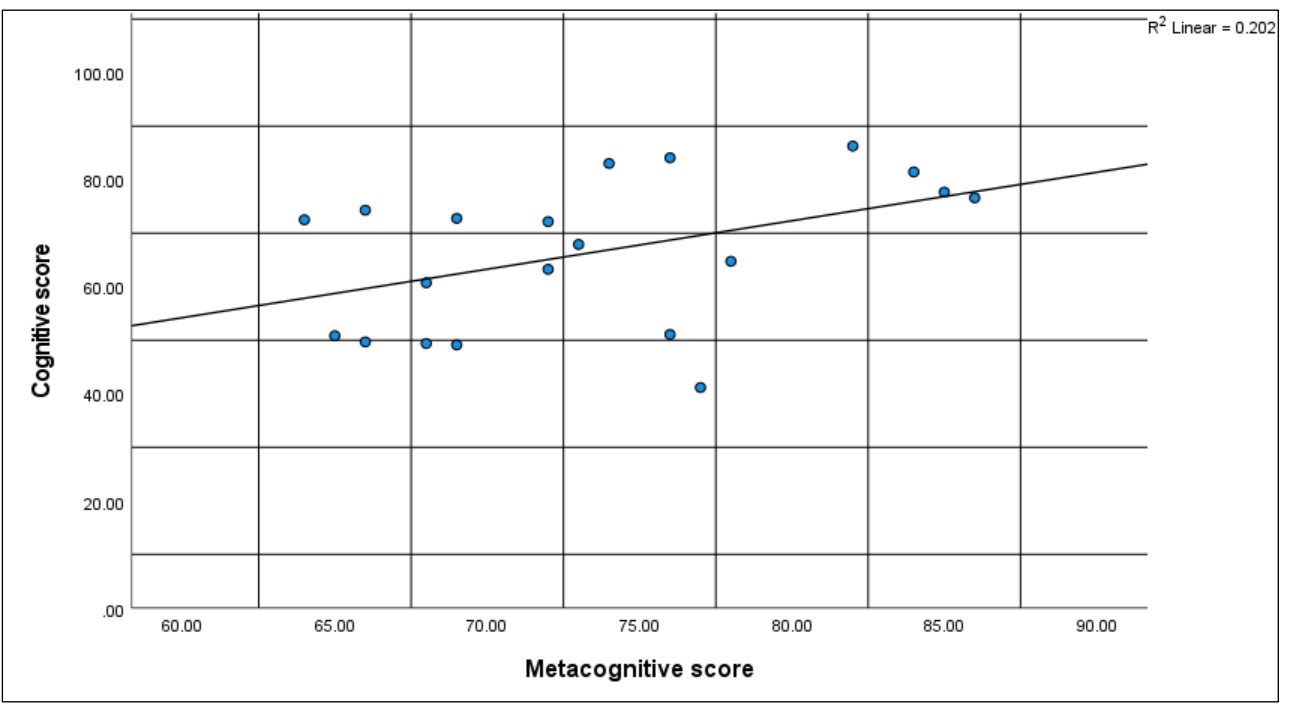

**Figure 6.** Scatter plot of cognitive scores by metacognitive scores.

## 5. Discussion

Processed data could be divided into four groups, as demonstrated in Figure 7. Group 1 includes students whose cognitive scores and metacognitive scores were the highest and positively corresponding. Next is Group 2, with Students 7, 8, 9, 10, and 11, whose cognitive and metacognitive scores were the second highest and positively correlated. Followed is Group 3, which includes Students 12, 13, and 14 whose metacognitive scores were higher but whose cognitive scores were lower than those in Group 2. Finally, Group 4, including Students 15, 16, 17, 18, 19, and 20, contains an apparent disagreement between cognitive and metacognitive scores.

Students 1, 2, 3, 4, 5, and 6 demonstrated diligent effort in all phases of the SRL cyclical model. Their log on the experiment system showed that what they had planned to do followed the task requirements. Furthermore, after planning, the students strictly followed their plan and executed it. They also demonstrated the achievement of high levels of revised Bloom's taxonomy by comparing different solutions and selecting the best one for solving the problem. Their answers to the MAI questionnaire also reflected the management aspect of their working progress.

Students 7, 8, 9, 10, and 11 achieved a rather good result in the accuracy of the prediction model. The log history showed that their work did not strictly follow the plan.

However, the performance of their work demonstrated a high level in revised Bloom's taxonomy scale.

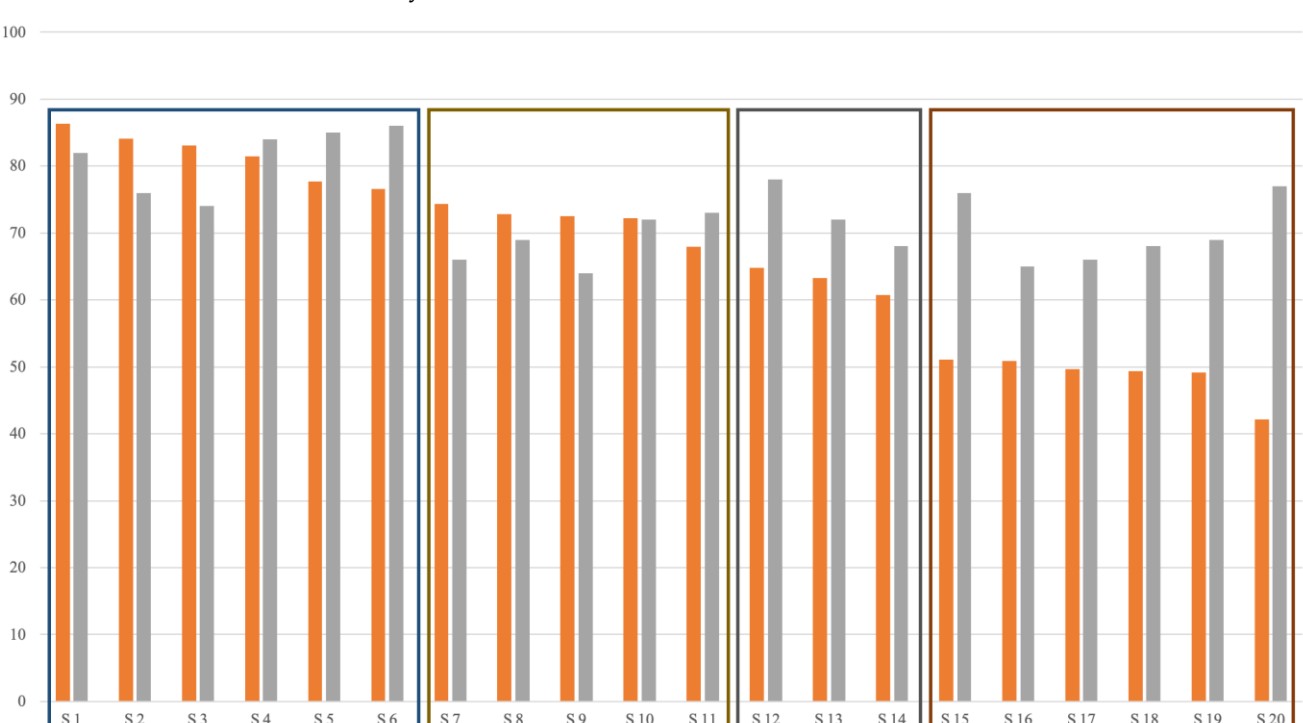

**Figure 7.** Groups of students according to types of correlation between cognitive and metacognitive scores.

Students 12, 13, and 14 planned reasonable solutions for the problem. They also executed their plan toward the final result. However, they did not pursue further performance to achieve a better outcome—a better prediction model accuracy—while these achievements were probably within their capacity.

In the case of Students 15, 16, 17, 18, 19, and 20, they were either students with a certain metacognitive maturity but lacking experience of machine learning courses or did not read the task diligently for comprehension; therefore, they demonstrated complicated planning and could not achieve the expected results.

From the data, it can be inferred that the students who lacked metacognitive strategies might have overestimated their ability to deal with complex tasks. They might not have considered the planning phase as important as the performing phase. There were also students who spent time planning before working on complex tasks but who did not perform as well as they had planned. It might be that they lacked either cognitive strategies or experiences with such a task.

SRL in online learning following the SRL cyclical model can be categorized into five profiles [28], namely:

- Non-self-regulators who learn with little or without applying necessary SRL phases;
- Forethought-endorsing self-regulators who carry out diligent activities in the planning phase of SRL;
- Performance/reflection-endorsing self-regulators who are active and demonstrate various cognitive activities in the performance phase of SRL;
- Super self-regulators who are active throughout all three phases of SRL;
- Competent self-regulators who are like super self-regulators and meanwhile adapt their progress and performance to meet their goals with an appropriate fit.

Applied to the SRL profiles, the groups above can be classified as follows. Group 1 can be considered as super self-regulators and competent self-regulators, Group 2 as



performance self-regulators, Group 3 as forethought-endorsing self-regulators, and Group 4 as non-self-regulators. One interesting observation is that the students in Group 4 may actually be self-regulators, who reach a rather high level in metacognition, yet they are not familiar with the type of the learning task in the experiment; hence they did not achieve high cognitive scores. Later, if they meet learning tasks of a similar kind in the future, this group of students, having gained experiences, could gradually reach Group 2 or 3.

To encourage and sustain SRL, designers of online courses should deliver learning tasks with concrete score-accumulative subtasks for planning and reflection purposes. In doing the subtasks, students perceive the cumulative contribution of their efforts to a goal rather than considering planning and reflection as unnecessary phases.

To each student group, the following instructional directions should sustain students' SRL ability.

Students in Group 1 showed a strong positive correlation between cognition and metacognition. Such a correlation might show that the students had experienced similar tasks and had accumulated cognitive strategies for working on such tasks and had planning habits. Therefore, when meeting the task, they made an appropriate plan and applied suitable methods for the task. The log history showed that they attempted the review quizzes sufficiently and gained high scores. Moreover, their results demonstrated positive relations between their plan and their implementation of the plan. To sustain the SRL ability of students in Group 1, course designers might provide tasks of similar complexity and a few more advanced tasks.

Students in Group 2 also showed a strong positive correlation between cognition and metacognition. It is noticeable that their cognitive and metacognitive scores were lower than Group 2, although they achieved rather good results in terms of the accuracy of the prediction model. Some of the accuracies were as good as those in Group 1. Their low cognitive and metacognitive scores implied a cursory plan or a hasty planning process. We could probably infer that the students in Group 2 had equivalent cognitive skills to Group 1 but lacked planning habits. The log history showed that they completed the review quizzes inadequately, resulting in low cognitive scores. To help students in Group 2 improve metacognitive strategies, course designers could provide additional review reports after a main task for the students to reflect on their performance.

Students in Group 3 might lack cognitive strategies preventing them from achieving better outcomes although they had planned a reasonable course of action. Students in Group 3 might need cognitive strategy training. Therefore, providing instructional tasks for training cognitive strategies would equip them with the requisites for complex tasks.

To support students in Group 4, course designers could assist the students in categorizing a learning task. In doing so, students could gradually link the task to others which they have experienced. In case the task is unfamiliar to the students, course designers could start from the task type into which students have categorized the task and then provide preparation tasks. Alternatively, the course designers could suggest other tasks close to the students' experiences.

In summary, this research demonstrates that the maturity level of a learner's SRL skills is measured through the measurement of SRL's two factors mentioned above: (1) the measurement of his or her cognition by revised Bloom's taxonomy framework, and (2) the measurement of his or her metacognition by the MAI questionnaire. The correspondence between these two factors is manifested via the maturity level of SRL skills. The relationship between cognition and metacognition in technology-enhanced self-regulated learning is that the stronger the relationship is, the more effectively and efficiently learners' SRL work. When they are working on complex learning tasks, the learners with high-level SRL are capable of:

- Establishing clear goals and objectives;
- Forming diligent plans to achieve their goals;
- Setting practically structured learning routine;
- Motivating themselves;

- Adapting their learning;
- Using a diversity of cognitive strategies.

Those capacities have been discovered and acceptable among research in SRL [1,29–33].

Learners are motivated when they can complete learning tasks through self-regulated learning. The more complicated a task is, the more learners are motivated when they complete it. Hence, providing complex tasks in a course can help strengthen the cognition and metacognition relationship of a learner. To encourage learners to self-regulate their learning on complex tasks, online course designers should pave the pathway to the complex tasks with preparation subtasks that are metacognitively or cognitively achievable and measurable for learners. Revised Bloom's taxonomy can be referred to in designing the tasks. These subtasks are opportunities and invitations for learners to practice metacognitive strategies and remember the cognitive strategies required for the main complex task. Performing such tasks is a concrete measurement to help learners know their cognitive and metacognitive levels. From a learner's perspective, when studying a subject or working on complex tasks, learners should not cut corners but instead should follow instructions if provided. Awareness of one's cognitive strategies repertoire, knowledge, and constraints before learning is a sign that one is metacognitive. The learning history of learners in TELEs is an informative source for evaluating learners' cognitive and metacognitive performance. Seeing cognitive and metacognitive progress would be helpful for learners to understand their SRL profile and a motivation for them to make adjustments to learning routines and persevere SRL.

For a visualization of cognitive level and metacognitive level as unit axes, we located the SRL profiles corresponding to cognitive and metacognitive levels in a diagram shown in Figure 8. Although the boundary between one profile and another is not concretely separated, we can see the specific character of each SRL profile in each quadrant of the diagram. The upper-right quadrant stands for learners with harmony between their cognition and metacognition; learners belonging to the lower-right quadrant engage in diligently metacognitive activities while those of the upper-left quadrant demonstrate cognitive activities; and those in the lower-left quadrant may know that they should improve their cognitive and metacognitive strategies and habits to benefit from self-studying. As learners transition their SRL profiles from the lower-left to the upper-right quadrant, we would see improvement in their SRL patterns such as planning more diligently with clear goals and objectives, having structured learning routines, learning with high motivation and adaptation, being able to apply a wide range of cognitive strategies to learning complex tasks and achieving high performance and results during learning. Learners themselves, through their own learning journey and experiences, can identify their current SRL profile. Advisors, when supervising their learners, can also help their learners identify their SRL profile. Moreover, that SRL profile can be used as a starting point to assist learners with cognitive or metacognitive improvement.

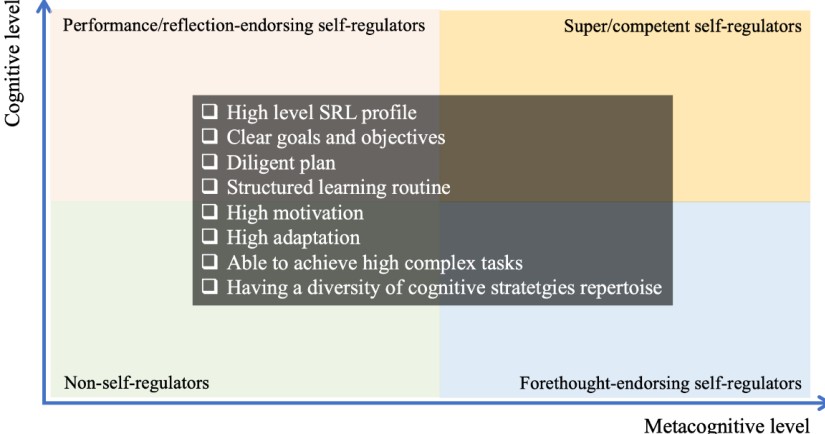

**Figure 8.** SRL profiles demonstrating the relationship between cognition and metacognition.

## 6. Conclusions

This research presented an empirical experiment to demonstrate the relationship between cognition and metacognition based on solid frameworks and measurements such as revised Bloom's taxonomy, the SRL cyclical model of Zimmerman, and Flavell's metacognition model. The precise relationship promises to enable TELEs to enhance and support learners' SRL ability adaptively and measurably during their learning process. Furthermore, the relationship also helps learners understand how much cognition and metacognition contribute to their SRL abilities and how their cognitive and metacognitive strategies operate. Hence, the learners can develop appropriate adjustments to their SRL skills when approaching future complex learning tasks. The experiment result illustrates that the relationship between cognition and metacognition in technology-enhanced SRL is verified and validated to a certain extent. With that result, further quantitative analysis into the relationship between cognition and metacognition is worth consideration.

This research has two main limitations: the experiment's sample size and the boundary between SRL profiles. First, the experiment was conducted on 20 participants only. It was a pilot study to verify the hypothesis from our proposed model. Although the experiment did categorize the SRL profiles as proposed by the hypothetical model, this small amount of test cases could not provide a statistical result that quantitatively supports the hypothesis. In addition, the experiment was about solving a complex problem in a short period of time, while a learner's SRL profile requires an extended amount of time to develop fully. Second, although the experiment demonstrated different SRL profiles, a separation between the SRL profiles was not clearly defined. Addressing these two limitations would qualitatively validate the hypothetical model about the relationship between cognition and metacognition in SRL and allow for the application of this model to improve SRL and SRL support.

With the results of this research, we recommend future work as follows. First, there should be research on detailed characteristics and features of each SRL profile. The motivation for conducting this research would be to help improve SRL so that learners could leverage their SRL abilities to achieve online learning successfully. Having SRL-profile-related characteristics would support the quantitative measurement and description of learners' SRL so that improvement methods could be recognized and applied. Once the SRL-profile-related features are formed, we recommend researching what course of action would help improve the SRL profile to higher levels.

With a deep understanding of SRL profiles and what can be done to improve them, not only will learners be able to understand their ways of learning but instructors will also have insights to design TELEs courses in a manner that assists learners in self-regulating their learning process.

**Author Contributions:** T.M.T. conducted the empirical study and composed the manuscript. S.H. provided counsel and revised and proofread the manuscript. All authors have read and agreed to the published version of the manuscript.

**Funding:** This research was supported by JSPS Grant-in-Aid for Scientific Research (B) 18H03346.

**Institutional Review Board Statement:** Not applicable.

**Informed Consent Statement:** Informed consent was obtained from all subjects involved in the study.

**Data Availability Statement:** The data presented in this study are available from the corresponding author upon reasonable request.

**Acknowledgments:** The authors would also like to thank the Japan Advanced Institute of Science and Technology (JAIST) and the Center for Innovative Distance Education and Research at JAIST for sponsoring this research and dedicating valuable time in counseling the authors to guide this research toward the right path.

**Conflicts of Interest:** The authors declare no conflict of interest.

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
