# Peer review of "An Empirical Study on the Relationship between Cognition and Metacognition in Technology-Enhanced Self-Regulated Learning"

_sustainability, doi:10.3390/su14073837_

Round 1

Reviewer 1 Report

Attached is a file with the minor revisions to be carried out.

Author Response

Thank you very much for dedicating time to reviewing the manuscript.
We appreciate your suggestion and corrections to our manuscript.
We would like to send our responses to your comments in the attachment.

Best regards,
Tuan Tran & Shinobu Hasegawa

Reviewer 2 Report

This paper is to present the student’s skill and self-regulated learning through two-hour assignment with on-line system. After the learning process from Figure 3 to Figure 11 then received a fruitful result.

This paper has provided an interesting topic, but fewer suggestions as below.

I do not know this paper belongs to information technology with application or sustainable learning, because I cannot find any information about sustainability.

In the section of method is lack of how many students were tested the proposed learning process.

The section 3.4 is long with many figures from Figure 3 to Figure 11. It is not necessary, because academic article is not a system guideline.

Please check the sentence between the Figure 6 and Figure 7. Which has an incorrect information such as “Figure 18Figure 18.”

Section 4, 5, and 6 is OK.

Appendix 1, 2, and 3 is not necessary. If need. Please check the Appendix 3, the Table 2 should be changed as Table 4.

Totally, this paper can be revised a better reversion.

Author Response

(The authors gave the same response as above.)

Reviewer 3 Report

First of all, thank you very much for inviting me to review the article entitled “An empirical study about the relationship between cognition and metacognition in technology-enhanced self-regulated learning”. The current paper described (through an experiment) how technology-enhanced learning environment are able to promote self-regulated learning (cognition and metacognition).

First, the special issue is about self-regulated learning, however, author should also link to sustained learning, more specifically, how the proposed revised Bloom’s taxonomy, SRL Cyclical model of Zimmerman, and Flavell’s metacognition model is able to help in creating a sustainable education environment and practice that endure. Would also be useful to provide practical pedagogical implications for practice (which is one of the goal of the special issue).

Some comments are as follows:

Is it necessary to mention “…at the 2019 International Conference on Frontiers in Education: Computer Science & Computer Engineering and was published in its proceedings…” typically, would just cite as author’s (2019) model….

Research objective is clearly stated.

In section 2.4 – the author/s discussed the model of a relationship between cognition and metacognition in technology-enhanced self-regulated learning, which is from a previous study of the authors? Paper should go through a similarity check.

Could be mistaken as the procedure for the current paper. This section is quite confusing. Is this the hypothesized model?

Method section – is this a 1 group experiment? Should be stated clearly (the design)

How did the authors take care of the individual differences?

Can discriminant analysis be used to separate the different type of learners? As noted there are 4 group of learners. (the four quadrants) are they mutually exclusive?

In sum, the paper is interesting with a detailed description on how study is accomplished. However, some parts are unclear and might be misleading, which make the paper harder to follow.

Author Response

(The authors gave the same response as above.)

Round 2

Reviewer 2 Report

This paper is my second time to read. The content and framework is better than the first version. 

After my double checked, this paper has corrected the problems, and response my suggestions.

The renewed version is suitable to be published on the journal. 

Reviewer 3 Report

Thank you for inviting me to review the revised version of the paper. as per reading the authors' response and going over the revised paper. the current version of the paper seems adequate for acceptance. This version is much clear and readers can easily follow on the procedure.